# The Potential Role of Adipose-Derived Stem Cells in Regeneration of Peripheral Nerves

**DOI:** 10.3390/neurolint17020023

**Published:** 2025-02-06

**Authors:** Sunil P. Mohan, Sivan P. Priya, Nada Tawfig, Vivek Padmanabhan, Rasha Babiker, Arunkumar Palaniappan, Srinivasan Prabhu, Nallan CSK Chaitanya, Muhammed Mustahsen Rahman, Md Sofiqul Islam

**Affiliations:** 1Department of Oral and Maxillofacial Pathology, Sree Anjaneya Institute of Dental Sciences, Kozhikode 673323, Kerala, India; 2Centre for Stem Cells and Regenerative Medicine, Malabar Medical College, Kozhikode 673315, Kerala, India; 3RAK College of Dental Sciences, RAK Medical and Health Sciences University, Ras AL Khaimah P.O. Box 12973, United Arab Emirates; nada.tawfig@rakmhsu.ac.ae (N.T.); vivek.padmanabhan@rakmhsu.ac.ae (V.P.); krishna.chytanya@rakmhsu.ac.ae (N.C.C.); mustahsen@rakmhsu.ac.ae (M.M.R.); sofiqul.islam@rakmhsu.ac.ae (M.S.I.); 4RAK College of Medical Sciences, RAK Medical and Health Sciences University, Ras AL Khaimah P.O. Box 11172, United Arab Emirates; rashababiker@rakmhsu.ac.ae; 5Human Organ Manufacturing Engineering (HOME) Lab., Centre for Biomaterials, Cellular and Molecular Theranostics (CBCMT), Vellore Institute of Technology, Vellore 632014, Tamil Nadu, India; arunkumar.p@vit.ac.in; 6Division of Phytochemistry and Drug Design, Department of Biosciences, Rajagiri College of Social Sciences, Cochin 683104, Kerala, India; prabhusbotany@gmail.com

**Keywords:** adipose tissue-derived stem cells, adipose-derived stromal cells, peripheral nerve injury, peripheral nerve repair and regeneration, regenerative medicine, nerve conduits

## Abstract

Peripheral nerve injuries are common complications in surgical and dental practices, often resulting in functional deficiencies and reduced quality of life. Current treatment choices, such as autografts, have limitations, including donor site morbidity and suboptimal outcomes. Adipose-derived stem cells (ADSCs) have shown assuring regenerative potential due to their accessibility, ease of harvesting and propagation, and multipotent properties. This review investigates the therapeutic potential of ADSCs in peripheral nerve regeneration, focusing on their use in bioengineered nerve conduits and supportive microenvironments. The analysis is constructed on published case reports, organized reviews, and clinical trials from Phase I to Phase III that investigate ADSCs in managing nerve injuries, emphasizing both peripheral and orofacial applications. The findings highlight the advantages of ADSCs in promoting nerve regeneration, including their secretion of angiogenic and neurotrophic factors, support for cellular persistence, and supplementing scaffold-based tissue repair. The regenerative capabilities of ADSCs in peripheral nerve injuries offer a novel approach to augmenting nerve repair and functional recovery. The accessibility of adipose tissue and the minimally invasive nature of ADSC harvesting further encourage its prospective application as an autologous cell source in regenerative medicine. Future research is needed to ascertain standardized protocols and optimize clinical outcomes, paving the way for ADSCs to become a mainstay in nerve regeneration.

## 1. Introduction

Peripheral nerve injury (PNI) is a common morbidity resulting in functional deficiencies encountered frequently in surgical and dental practices. In routine surgical management, autologous grafting is widely practiced as the treatment of choice; however, it has notable limitations, including donor site morbidity and healing with scar formation, often leading to suboptimal outcomes. PNI occurs in 1.6% to 2.6% of trauma cases, primarily affecting young adult males aged 21 to 40 years old [1,2]. Recently, tissue engineering for nerve repair has gained momentum, utilizing nerve conduits, neurotrophic factors, and stem cells to facilitate regeneration [3]. A thorough understanding of the physiological events involved in nerve regeneration after PNI has paved the way for advancements in nerve tissue engineering, particularly through stem cells. Peripheral nerves possess a significant regenerative capability. The implementation of stem cells during the repair can accelerate the regenerative process, thereby facilitating a swifter restoration of functionality [4,5] (Table 1).

Table 1: Comparison of various treatment approaches for peripheral nerve repair, highlighting their descriptions, advantages, limitations, and clinical applications. Autologous nerve grafting remains the gold standard for nerve gaps less than 4 cm, while stem cell therapy using ADSCs (adipose-derived stem cells) offers promising regenerative potential through paracrine effects. Synthetic and natural nerve conduits provide structural guidance for nerve regeneration, with biodegradable natural conduits excelling in biocompatibility but limited by structural strength. Each approach is tailored to specific injury types and gap lengths.

Since the discovery of adipose-derived stem cells (ADSCs) by Zuk PA from processed lipoaspirate cells, these cells have shown potential for differentiating into various lineages, including osteogenic, chondrogenic, myogenic, and neurogenic tissues [10]. ADSCs are among the most accessible sources of stem cells in the human body, making them suitable for both autologous and allogeneic regenerative applications. The cellular composition of adipose tissue, which includes pericytes, adipocytes, and fibroblasts, facilitates easy isolation and immediate transplantation upon harvesting [11,12]. Although the molecular mechanisms behind nerve regeneration are well-studied, current research aims to uncover further insights into complex processes, such as cellular histopathological details and synaptic plasticity, which could enhance the complete regeneration of damaged nerves, including those with degenerative neurological issues [12]. This review aims to deepen our understanding of PNI, the role of ADSCs in nerve regeneration, and their therapeutic applications to improve clinical outcomes [4].

## 2. Methodology

This comprehensive review adopts a systematic and integrative approach to synthesizing and analyzing the existing literature on the role of ADSCs in peripheral nerve repair and regeneration. The methodology involves a rigorous process, ensuring the inclusion of high-quality and relevant studies to produce reliable and clinically significant insights. The literature search was performed across multiple scientific databases, including PubMed, Scopus, and Web of Science. Keywords such as “adipose tissue-derived stem cells”, “peripheral nerve injury”, “peripheral nerve regeneration”, “nerve conduits”, and “regenerative medicine” were utilized, along with Boolean operators to refine the search. The inclusion criteria were confined to peer-reviewed articles, case studies, systematic reviews, and clinical trials published between 2000 and 2024. Non-peer-reviewed publications, irrelevant studies, and those lacking methodological detail were excluded. Reference lists of included articles were examined for additional studies. Extracted data were analyzed for accuracy and comprehensiveness.

## 3. The Pathophysiological Cascade Following Peripheral Nerve Injury

### 3.1. Classification of Peripheral Nerve Injury

Seddon initially classified PNI and later reclassified it by Sunderland [13,14]. Seddon categorizes injuries into neurapraxia, axonotmesis, and neurotmesis, while Sunderland divides them into five grades. Grade 1 of Sunderland’s classification, which is the same as Seddon’s neuropraxia classification, states that the mildest type of nerve damage affects only the conduction of the axon and causes segmental damage to the myelin sheath (Figure 1). Usually, nerve fibers lose conduction when they are directly compressed or stretched, or when there is significant ischemia in the area. However, the nerve fibers remain continuous with the undamaged endoneurium, perineurium, and epineurium. Recovery is often rapid due to limited cell damage and surrounding tissue supporting regeneration. Wallerian degeneration does not occur at this level of injury. When axons and their myelin sheath are severed from the main nerve fiber, they break into smaller pieces. This process is called Wallerian degeneration. The damaged axon stimulates Schwann cells and macrophages to remove the damaged axons, myelin, and other cellular debris.

Sunderland’s Grade II injuries involve damage solely to the axon and the myelin damage. Grade III injuries encompass myelin, axon, and endoneuria damage. Grade IV injuries extend to damage to the myelin, axon, endoneurium, and perineurium. The Seddon classification matches Grade II–VI injuries with axonotmesis (Figure 1). Axonotmesis causes Wallerian degeneration on the axon near and far from the injury site. In Seddon’s classification, Sunderland’s Grade V is equivalent to neurotmesis, where there is a complete disconnection of the peripheral nerve ending that includes all the supportive structures [15]. Figure 1 illustrates the microanatomical and structural changes during PNI at each stage.

In cases of neurotmesis, direct surgical intervention is required to approximate the nerve endings, often with assistance from Schwann cells to support regeneration [5,16,17]. During traction injuries, nerves undergo tensile stress along their collagen fibers. The perineurium, being more elastic than the endoneurium, is less susceptible to tearing [18]. When exposed to high traction forces, nerve root avulsion can occur, disrupting the nerve’s anatomical and physiological functions and leading to functional loss at the end organ. Injuries caused by sharp objects like knives, glass, or surgical procedures can result in partial or complete segmental damage to the nerve, with mild lacerations potentially allowing for natural regeneration [19]. In cases of crush or compression injuries, early alleviation of pressure on the affected nerve, ideally within 8 h of the incident, can facilitate the restoration of normal function in the surrounding microvascular plexus. Generally, the peripheral regions of the nerve suffer greater damage compared to the central areas in such injuries. This is due to the ischemic forces that predominantly impact the peripheral vasculature. Other minor causes of PNI include electrocution, thermal, vibration, and percussion injuries [19]. The outcome of nerve injury largely depends on its severity [14]. Peripheral nerves are composed of several fascicles, not just a single nerve fiber. This fascicular arrangement demands prolonged support during the regeneration process. Beyond other peripheral nerves, oral nerves, including the inferior alveolar and mental nerves, are vulnerable to various types of traumas. This is particularly prevalent during prolonged dental procedures, maxillofacial surgery necessitated by accidents, and orthognathic corrective surgeries. These nerves share similar regenerative challenges, highlighting the importance of exploring effective regenerative strategies. ADSCs could offer a potential solution for supporting regeneration in both PNI, aiding in functional restoration through their neurotrophic and angiogenic capabilities [4,11].

### 3.2. Cellular and Molecular Events Following Peripheral Nerve Injury

PNI initiates a series of molecular and cellular events, divided into acute injury response with Wallerian degeneration and inflammation continued by regeneration stages, to promote healing. Many cells and their products are actively involved in PNI and regenerative processes. Before discussing the molecular and cellular events following injury, it is imperative to understand certain ubiquitous properties of nervous tissue. Unlike the renewable epithelial cells that regenerate endlessly, human nerve cells are permanent cells and do not regenerate continually. Neuronal cell bodies (soma or perikarya) are found in the CNS and are vital for renewal and repair. The PNS consists mainly of axons (neuronal extensions), with the exception of peripheral ganglions, which contain neurons outside the CNS. Peripheral nerve health and regeneration are dependent on CNS health, as PNS axons originate in the CNS. Furthermore, the less-explored Blood-Nerve Barrier (BNB) system, which is similar to the Blood–Brain Barrier (BBB) and Blood-Spinal Cord Barrier (BSCB), demonstrates comparable endothelial tight junctions, necessitates further exploration to fully explain the pathophysiological and molecular details after PNI for effective treatment [20].

Most CNS neuron development occurs before age five, but PNS development remains until maturity, with axon lengthening and the addition of Schwann cells and other support structures. The continuing growth of nerve fibers is influenced by genetic, environmental, and epigenetic factors, but the underlying molecular mechanisms and cellular biology necessitate further research. Axons in the PNS are meticulously supported by a multi-layered supportive framework up by Schwann cells forming myelin sheath, then endoneurium for individual axons, perineurium for groups of axons, and finally epineurium for many groups of axons (Figure 1). Axons, extensions of soma residing in the CNS, may stretch for distances up to a meter in the PNS. In myelinated nerves, many Schwann cells wrap around individual axons, leaving distinct gaps of 0.2 mm to 1.5 mm known as notes of Ranvier, measuring a few microns; whereas, in the unmyelinated nerves, a single Schwann cell supports multiple axons named the Remak bundle. Notably, specific sensory nerves, especially those responsible for pain transmission, and autonomic nerves, such as sympathetic postganglionic fibers, are unmyelinated. The repair ability of these unmyelinated nerves is less than that of the myelinated nerves and needs more support during regeneration. Peripheral nerves consist of many fascicles, rather than a singular nerve fiber. The sciatic nerve, the largest in mammals, contains 140 fascicles in the gluteal region and 50–80 in the mid-thigh of adult humans [21,22]. Due to this fascicular arrangement, sustained support is necessary during the regeneration. While the complete regeneration of nerve cells after injury remains debatable, the repair is recognized if the perikarya and surrounding support cells at the axonal level actively participate in the process.

Understanding how myelin is repaired is key to developing regenerative treatments for axon-related disorders. Myelin sheath damage is linked to many neurological diseases, such as multiple sclerosis, Alzheimer’s disease, encephalomyelitis, and neuropathy [23]. Nerve regeneration is complex and requires more than just axon regrowth; it involves all nerve structures for comprehensive functional and end-organ recovery. The ability to heal after an axonal injury is inversely related to the distance from the cell body. In addition, the duration and intensity of the insult, the age of the patient, and associated inflammatory responses may affect repair and regeneration after PNI. Repair in the PNS is functionally successful only if the axon regenerates, depending on signals from the cell body in the CNS (or rarely PNS for peripheral neurons). However, the surrounding structures of the axon (the myelin sheath, epineurium, endoneurium, and perineurium) can regenerate. This is because these structures originate from the Schwann cells and fibroblasts, which are renewing cells.

### 3.3. Acute Injury Response and Wallerian Degeneration

In response to PNI, axonal and myelin fragmentation occurs in a proximo-distal direction [24,25]. Axonal damage leads to a calcium influx, activating enzymes that degrade axons and myelin. In the CNS, neuronal cytoskeletal changes include chromatolysis, characterized by cell body enlargement, eccentric nuclear positioning, and dispersed Nissl granules (Figure 2). Following axonal injury, the neuronal cell body downregulates genes responsible for nerve signal transmission while temporarily upregulating those involved in axonal regeneration, reflecting a decline in regenerative capacity over time. Wallerian degeneration begins when the axon disconnects from the neuronal cell body, leading to a localized accumulation of debris that hinders repair.

Schwann cells play a vital role by proliferating and undergoing morphological adaptations, allowing them to function as scavenger cells that clear debris through recruitment and coordination with macrophages. These processes occur in the proximal nerve stump under the influence of Schwann cells, which also facilitate bridging the injury gap. Additionally, Schwann cells secrete mRNA and signaling molecules such as P75, monocyte chemoattractant protein-1, and nerve-derived growth factors to promote repair [26]. To further support regeneration, Schwann cells establish an extracellular matrix (ECM) and release neurotrophic factors, including Glial-Derived Neurotrophic Factor (GDNF) and Brain-Derived Neurotrophic Factor (BDNF), which stimulate axonal growth. Wallerian degeneration is evident from Grade II injuries, and its severity increases as the injury progresses. In cases of severe nerve injury, apoptosis may be triggered.

### 3.4. Axon Regeneration

The regenerative process within the injury gap can be divided into five distinct phases: the fluid phase, the matrix phase, the cellular migration phase, the axonal phase, and the myelination phase. Initially, during the fluid phase, the gap at the injury site is filled with ECM precursors laden with neurotrophic factors. Subsequently, during the matrix phase, these ECM precursors assemble into an acellular fibrin scaffold that facilitates regeneration. The cellular migration phase is characterized by the migration of Schwann cells and some endothelial cells and fibroblasts across this fibrin bridge. The migrated Schwann cells align to form bands of Büngner, which guide axonal sprouting at the injured end and facilitate reinnervation to restore end-organ function. Lack of this guidance is correlated with neuroma formation [27]. After axonal regeneration, Schwann cells revert to their original myelinating and proliferative phenotype, consequently contributing to functional repair and maintenance. This regeneration process, nevertheless, is limited by the length of the transection gap of approximately 1.5 cm in rats and 4 cm in humans [28,29]. Axon regeneration happens at a rate of 1–3 mm per day; consequently, an injury spanning 60 cm would require nearly two years for complete regeneration [30]. This process is contingent upon the sustained presence of adequate perfusion, nutrients, and cellular support. In order for comprehensive regeneration to occur, the presence of cells that aid in nerve fiber regeneration is important. These cells include Schwann cells, endothelial cells, pericytes, immune cells, and fibroblasts. ADSCs can deliver this necessary, comprehensive cellular support, which is the focus of this review. During PNI, Schwann cells and other glial cells promote regeneration through several mechanisms. Additionally, Schwann cells assist in clearing degraded axons and myelin sheaths, which further support the regenerative process [31,32].

### 3.5. Adipose-Derived Mesenchymal Stem Cells (ADSCs)

ADSCs are a multipotent regenerative resource derived from adipose tissue consisting of various cell types, including mature adipocytes and stromal vascular fraction (SVF) cells (Figure 3). The SVF provides a rich amount of ADSCs, which can be easily isolated and processed for therapeutic use. SVF cells are a heterogeneous mixture of pericytes, fibroblasts, endothelial cells, smooth muscle cells, mast cells, and pre-adipocytes. When propagated under prevailing conditions, these cells form a relatively homogeneous population of ADSCs, distinguished by their regenerative abilities and denoted as adult stem cells (ASCs) [32,33]. The International Society for Cellular Therapy (ISCT) has described minimal standards for human Mesenchymal Stem Cells (MSCs) [34]. According to these guidelines, MSCs must adhere to plastic surfaces in standard culture, possess the capacity to differentiate into osteogenic, adipogenic, chondrogenic, and various other lineages, and express specific markers such as CD73, CD90, and CD105 while lacking hematopoietic markers like C-kit, CD14, CD19, CD34, CD45, CD77, and human leukocyte antigen-DR. ADSCs meet nearly all ISCT criteria for MSCs, making them highly appealing for regenerative medicine. Ideally, stem cells for therapeutic applications should be easily accessible, capable of rapid and controlled proliferation, possess immune-modulatory properties, and demonstrate robust survival in vivo to support relevant tissue formation [35]. Unlike embryonic stem cells, which are totipotent but face ethical concerns due to the need for embryo extraction, ADSCs offer a multilineage potential that allows differentiation predominantly into mesodermal tissues, as well as some endodermal and ectodermal structures [10,36].

ADSCs share similar protein expression profiles and differentiation abilities with bone marrow-derived stem cells (BMSCs) [4,35]. A significant advantage of ADSCs is their high yield from minimally invasive harvesting procedures, with approximately 3.5 × 10^5^ to 1 × 10^6^ cells obtainable per gram of aspirated fat tissue [36,37]. Originating from embryonic mesenchyme, adipose tissue has a multilineage capacity, enabling the formation of mesodermal structures like bone and muscle, as well as ectodermal derivatives such as neural tissues [4,33]. Processed lipoaspirate (PLA) preserves mesenchymal properties and demonstrates a tendency for neurogenic lineage, emphasizing its potential for nerve regeneration [38,39]. Since their identification in 2001, several methods have been established for ADSCs isolation, with the collagenase digestion method being the most popular, which involves treating adipose tissue with collagenase enzyme solution, followed by washing with phosphate-buffered saline to eliminate red blood cells. After removing the infranatant, the solution is centrifuged to make SVF pellets, which can be further propagated to produce ADSCs colonies [40]. ADSCs are specifically promising for peripheral nerve regeneration due to their neurotrophic and angiogenic factor secretion, as well as their capability to support ECM production. Their accessibility, ease of isolation, and regenerative potential make them a transformative approach to enhancing nerve repair and functional recovery in various clinical settings, including dental and maxillofacial applications [41,42] (Table 2).

Table 2: Key properties and functions of adipose-derived stem cells (ADSCs) in nerve regeneration. ADSCs provide high cell yields with minimal invasiveness, making them scalable for therapeutic applications. They secrete a variety of growth factors, including VEGF, EGF, and neurotrophic factors like BDNF, which promote angiogenesis and neuroprotection. With the ability to differentiate into Schwann-like and neural cells, ADSCs directly support nerve repair. Their paracrine signaling enhances the local microenvironment, while their immunomodulatory properties reduce inflammation and minimize tissue damage, collectively making ADSCs a promising candidate for peripheral nerve regeneration.

### 3.6. Angiogenic and Neurotrophic Factors in ADSCS

Vascular Endothelial Growth Factor (VEGF) is a key regulator of angiogenesis, which is crucial for tissue regeneration. ADSCs are known to secrete high levels of VEGF, especially under hypoxic conditions, which assists the formation of new blood vessels essential for tissue repair and regeneration [47,48]. Additionally, ADSCs secrete other potent growth factors, including Angiopoietin-1, epidermal growth factor (EGF), insulin-like growth factor-1 (IGF-1), fibroblast growth factor 2 (FGF-2), platelet-derived growth factor (PDGF), Transforming Growth Factor-beta (TGF-β), and Hepatocyte Growth Factor (HGF), which are important for the repair and regeneration of PNI (Figure 3) (Table 2) [7,12,32]. This angiogenic capacity of ADSCs is achieved by direct differentiation into endothelial cells and paracrine effects, whereby they release signaling factors that stimulate adjacent cells to enhance vascularization [47,48]. In addition to their role in stimulating angiogenesis, ADSCs preserve neurons by promoting cellular survival and repair, and by releasing neurotrophic factors that avert cell apoptosis. Since the surviving neuron is important for axon regeneration, the neuroprotective ability positions the ADSCs as a valuable asset in the realm of nerve regeneration [7,12,49]. Furthermore, ADSCs can form neurospheres (clusters of neural stem cells) that differentiate into Schwann cell-like cells capable of generating myelin and supporting neurite growth. These neurospheres can be stimulated to develop into glial-like cells, neural cells, and Schwann cells, which are vital for nerve regeneration [50,51]. Even in a non-targeted approach, ADSCs’ intrinsic paracrine effects play a promising role in the transformation of resident Schwann cells. The ADSCs extend to assist the local Schwann cells at the injury site by promoting their proliferation and function. This synergistic interaction is likely due to neurotrophic factors produced by ADSCs, creating a retrieving milieu that accelerates nerve regeneration [12,31]. With their distinctive combination of angiogenic, neuroprotective, and differentiation capabilities to develop myelin-producing Schwann cells, ADSCs are promising sources for regenerative applications in PNI and functional recovery (Figure 3).

### 3.7. Nerve Conduit

Nerve conduits (NCs) serve as vital guides for supporting nerve regeneration in cases of PNI [50,51]. Nerve conduits assist regeneration across injured gaps, especially when the injury distance is 20–25 mm. Regenerative outcomes increase when the conduits are treated with cellular elements and have a distinct porosity that matches the biological design of the nerve, improving integration and functional recovery [3,50]. Nerve conduits can be prepared from synthetic or natural materials. Synthetic biodegradable polymers for nerve conduits include polyurethanes like polyesters such as PGA (poly (glycolic acid)), PCNU (polycarbonate urethane), and PEUU (poly (ester urethane) urea); PLA (poly (lactic acid)), PLGA (poly (lactic acid-co-glycolic acid)), and PCL (polycaprolactone). Nonbiodegradable selections are silicone, poly (tetrafluoroethylene), and polystyrene [14]. Silicon-based conduits are commonly used synthetic material due to its durability; however, it has significant limitations, silicon-based conduits often require secondary surgery for removal, as they may cause adverse reactions or rejection if left in the body for extended periods, making them less ideal for long-term use. In contrast, natural conduits are often preferred as they are biodegradable, have intrinsic cell-binding potential, and possess molecular adhesion properties compatible with other tissues. Among the naturally sourced materials, collagen, silk, chitosan, gelatin, keratin, hyaluronic acid, agarose, fibrin, and alginate have been extensively studied, either individually or in combination with other natural or synthetic polymers [50]. Additionally, natural conduits generate fewer by-products during biodegradation, reducing the risk of pH imbalance and enhancing biocompatibility [25,35,47,50] (Figure 4).

Emerging polymers, such as polypyrrole, offer exciting possibilities for nerve repair due to their conductive properties and capacity to support nerve regeneration [52]. Polypyrrole-based conduits are particularly beneficial for regenerating nerves in electrically active environments, potentially enhancing neural signaling and repair. However, when nerve injuries exceed 3 cm, conduits alone may be insufficient. For these long-gap injuries, additional supportive factors, such as ADSCs or neurotrophic factors, are needed to augment the regenerative potential. Integrating nerve conduits with ADSCs, which secrete neurotrophic and angiogenic factors, significantly improves the chance of successful nerve regeneration even in challenging cases involving extensive nerve gaps.

The integration of nerve conduits with regenerative cells like ADSCs opens new avenues for treating complex nerve injuries. This approach leverages the structural support of conduits alongside the biological enhancement provided by stem cells, positioning this combination as a transformative strategy for nerve repair and functional recovery in diverse clinical applications.

### 3.8. Role of Scaffold and Matrix Constituents in Nerve Regeneration for ADSCs

Creating an appropriate microenvironment is paramount for successful nerve regeneration, especially when leveraging ADSCs for nerve repair. The scaffolds or conduits used must support ADSCs survival, proliferation, and paracrine signaling to maximize regenerative outcomes [32,53,54]. Hydrogels, which structurally resemble the ECM with their three-dimensional cross-linked networks, are widely utilized in regenerative medicine due to their favorable properties. Hydrogels are highly biocompatible, elicit minimal inflammatory responses, and cause less tissue damage upon implantation. They facilitate excellent nutrient and metabolite exchange, maintaining a supportive framework for cellular survival and function [53,54]. One promising hydrogel is N-methacrylate glycol chitosan (MGC), which is highly soluble and can be conveniently injected into the desired site. MGC can be cross-linked using visible or ultraviolet light, forming a stable scaffold that supports ADSCs engraftment and activity [50,54]. Alginate gel, a natural polysaccharide that rapidly cross-links in the presence of calcium ions, is another FDA-approved hydrogel commonly used as a wound dressing. It is biodegradable, porous, and injectable, making it apt for regenerative applications [55]. Alginate gels can form uniform structures and do not have adverse effects on ADSCs’ morphology, viability, or regenerative potential, but may not fully simulate the intricacy of the native ECM. Hyaluronic acid is another hydrogel compatible with ADSCs, modeling ease of culture and integration [56]. Its natural occurrence in the ECM and ability to support cell migration and proliferation make it a suitable scaffold material for nerve regeneration. Polyethylene glycol (PEG) is an advanced synthetic hydrogel that permits the free diffusion of nutrients and metabolites due to its similarity to the ECM. PEG hydrogels offer excellent biocompatibility and can be engineered with particular mechanical properties appropriate for nerve tissue engineering [57]. In peripheral nerve regeneration, PEG-based hydrogels can serve as effective delivery vehicles for ADSCs, promoting tissue repair and functional recovery. The utilization of these hydrogels as scaffolds in combination with ADSCs can establish a supportive milieu that may augment nerve regeneration. The scaffolds furnish structural assistance and direct cellular growth, while ADSCs contribute through differentiation and the secretion of neurotrophic factors. This synergistic approach holds substantial promise for advancing regenerative therapies aimed at restoring function in damaged peripheral nerves, eventually improving patient outcomes (Figure 4) (Table 3).

Table 3: Comparison of materials used in nerve repair scaffolds and their compatibility with adipose-derived stem cells (ADSCs). N-methacrylate glycol chitosan (MGC) is injectable and forms stable cross-linked structures, although it requires UV or visible light for cross-linking. Alginate gel, a natural polysaccharide, supports ADSCs viability and is FDA-approved but forms homogeneous structures. Hyaluronic acid enhances cell migration and is highly biocompatible, although it may degrade rapidly in certain applications. Polyethylene glycol (PEG), a synthetic hydrogel, mimics extracellular matrix properties and supports nutrient transfer but often requires chemical modification for specific nerve repair needs. Each material offers distinct advantages and challenges, making them suitable for different regenerative strategies.

## 4. Discussion

Regenerative medicine’s objective is to substitute damaged tissue with functional tissue. Cellular injury, which can result from factors including trauma, infection, genetic predisposition, and degenerative diseases, impacts all four primary cell types present throughout the body (epithelial, connective, muscle, and nervous tissue), and regeneration necessitates support for all associated tissues. Nerve tissue possesses limited regenerative capacity and requires additional support. Stem cell therapy has surfaced as an assuring tool for nerve regeneration. ADSCs meet most of the criteria set by the ISCT, making them an appealing choice for regenerative applications. ADSCs can differentiate into both mesenchymal and non-mesenchymal cell lineages, which can assist in regeneration. Among MSCs, ADSCs have shown particular promise due to their ease of harvest, simple culture conditions, substantial cell yield and high survival, ethical safeness, and ability to secrete essential growth factors that support nerve regeneration [35]. Adipose tissue is abundantly distributed in the human body and can be obtained through liposuction under local anesthesia and causes less trauma than BMSCs. ADSCs show a high degree of safety, as they do not induce tumorigenesis and teratogenesis [35]. Additionally, they demonstrate compatibility with many biomaterials and, when combined with scaffolds during tissue engineering, can promote tissue regeneration [62].

Nerve tissue engineering (NTE) holds immense potential in regenerative medicine as the need for innovative treatments for nerve injuries remains to rise. Conventional approaches, such as autologous nerve grafting, have long been the gold standard for restoring nerve defects. However, despite their efficacy, these methods have significant limits, including donor site morbidity, imperfect recovery, and issues with size discrepancy between donor and recipient tissues [27]. Autologous graft tissues like muscles and veins can cover small to moderate defects (0.5 to 4.0 cm) and partly reestablish motor and sensory function, but they often fall short in more widespread injuries [63]. Previous studies have demonstrated that ADSCs secrete angiogenic and neurotrophic factors, such as BDNF, NGF, and GDNF, all of which have shown positive effects on peripheral nerve repair [3,64]. Researchers have also noted that combining BDNF with gelatin tricalcium phosphate (a biomaterial used to support tissue growth) yields promising results, indicating the potential for synergistic approaches in NTE [3,4,11].

In an experimental study by Kilic et al., autologous fat tissue exhibited the highest isometric titanic force recovery in nerve repair. The study confirmed a significant increase in axon count, nerve density, and myelin thickness, suggesting that whole adipose tissue can act as a natural barrier, decreasing adhesion and fibrosis, and supporting an environment favorable to nerve regeneration [65]. Furthermore, findings on uncultured undifferentiated ADSCs (uuADSCs) have shown that these cells can assist axonal regeneration and re-innervation in nerve conduits more effectively than saline-filled controls, as demonstrated by the work of Suganuma et al. [66]. The use of cultured ADSCs, expanded over 2–12 passages to increase their quantity and enhance neurotrophic factor secretion, has further improved the regenerative potential of ADSCs. Cultured undifferentiated ADSCs (cuADSCs) have verified up to 80% regenerative capacity in PNI, mainly in experiments on the cavernous nerve in rats. Key parameters, such as neuronal nitric oxide synthase expression, myelinated axon development, and an optimal muscle-to-collagen ratio, propose that ADSCs provide substantial support for histological and functional regeneration. Several studies have recognized the ability of ADSCs to regenerate sciatic and facial nerves, highlighting their utility in repairing diverse nerve types [4,32,66].

Specifically, ADSCs have shown efficacy in facial nerve regeneration, with experimental groups displaying improved nerve conduction and functional recovery, such as vibrissae movement and enhanced motor function in walking track assessments [4,12,32,40,67]. ADSCs have been found to survive up to 12 weeks after transplantation into injured peripheral nerve tissue, indicating their resilience in vivo [68]. Interestingly, successful nerve repair does not always require full differentiation of ADSCs into neural lineage cells. Studies suggest that the neurotrophic factors and ECM provided by ADSCs play a pivotal role in facilitating repair, whether through direct differentiation or paracrine effects [4,67]. ADSCs can enhance wound healing, lymphangiogenesis, angiogenesis, modulate ECM, and minimize fibrosis during repair [69,70,71,72]. Human platelet lysate (HPL) is an effective alternative to fetal calf serum (FCS) for the propagation of stem cells. HPL derived from outdated platelets has been proposed as an effective and reliable source to expand the ADSCs and significantly improve the neurotrophic potency [73,74]. The use of autologous adipose tissue and autologous platelet lysate to propagate the cells delivers the possibility of precision medicine. This approach can reduce the risk of viral and prion transmission associated with the use of growth serum derived from animals and reduce necessity for the animal sacrifice.

Before translating these findings into clinical applications, adherence to Good Manufacturing Practices (GMP) is essential for producing clinical-grade human SVF and ADSCs. Regulatory bodies, including the FDA and other national agencies, have established guidelines to ensure the safe production of adult stem cells, categorized as minimally manipulated and largely autologous. These stringent requirements aim to minimize the risk of contamination and include routine testing such as endotoxin assays. With the increasing demand for minimally manipulated ADSCs in surgical applications, companies like Cyton Therapeutics (San Diego, CA, USA) have developed devices capable of rapidly processing lipoaspirate to separate adipocytes and SVF, thus facilitating the efficient and compliant use of ADSCs [39,75,76].

Further enhancing the promise of ADSCs in NTE, 3D bioprinting technology has enabled the fabrication of nerve conduits that closely mimic natural nerve structures. By integrating ADSCs with 3D-printed nerve conduits, it is possible to create biologically active scaffolds that provide structural support and release bioactive molecules essential for nerve regeneration. As 3D bioprinting technology progresses, nerve conduits can be engineered to include ECM constituents, neurotrophic factors, and even electrical conductivity, making them an increasingly viable substitute to traditional autografts and synthetic conduits [50,77,78]. A clinical trial is validating the effectiveness of ADSCs combined with a human amniotic membrane in treating brachial plexus injuries (NCT04654286). Furthermore, phase 1 clinical trials are currently underway to investigate the potential of ADSCs in the treatment of facial nerve injuries (NCT02853942, NCT04346680). ADSCs offer an exceptional and versatile approach to nerve regeneration, owing to their ability to promote angiogenesis, release neurotrophic factors, and support cellular scaffolding. Combined with advancements in scaffold design, bioprinting, and regulatory compliance, ADSCs represent a revolutionary development in regenerative medicine with the potential to reestablish function, sensation, and quality of life for patients. As research progresses, the integration of ADSCs, nerve conduits, and bioprinting technologies guarantees to reshape the landscape of NTE, spreading the way for more applicable and accessible treatments for PNI.

### 4.1. Repurposing Medically Wasted Tissues for Nerve Regeneration

The innovative approach of repurposing medical waste from cosmetic procedures (liposuction) for nerve regeneration research aligns with current advancements in regenerative medicine. Schwann cells play a crucial role in nerve injury by proliferating, dedifferentiating, and developing Büngner bands that direct regenerating axons. Neural crest, a unique cell population gives rise to diverse cell types, including Schwann cells and satellite cells, which are vital for the sustenance and maintenance of peripheral nerves. During embryogenesis, cells from the neural crest significantly influence the development of the PNS in vertebrates [79]. Satellite cells are found in the peripheral ganglia border and support the neuronal cell bodies, maintaining their microenvironment and moderating their activity. The use of neural crest-derived stem cells, including dental pulp stem cells (DPSCs), offers significant potential for peripheral nerve regeneration [78,80]. Teeth removed for orthodontic reasons due to overcrowding are often discarded as medical waste. However, these teeth can be a valuable source of DPSCs. They express multiple factors that support neuronal and axonal regeneration, including neurotropic factors like BDNF and GDN [81]. This suggests a potential research direction for investigating the merged use of ADSCs and DPSCs for peripheral nerve regeneration.

Adipose tissues throughout the body have distinctive developmental origins. Notably, head and neck adipose tissue can originate from the neuroectoderm, persuading its functional characteristics compared to adipose tissue in other regions. Research using tissue-labeling techniques on adipose tissue from the cephalic region proposes that this tissue originates from the neural crest [82]. This finding opens up a new prospect for using adipose tissue from the orofacial region to repair nerve injuries. Stem cells of dental pulp, which originate from neural crest cells, have shown potential in neural differentiation. Further research is essential to explore the potential of orofacial adipose tissue in neural regeneration. Importantly, all tissues in the head and neck region also arise from neural crest cells, signifying plasticity and the ability to produce diverse connective tissues, including the adipose tissue. This highlights the potential of applying tissues obtained from cosmetic procedures like liposuction for neural regeneration and differentiation therapies in the context of facial injuries. Their ability to differentiate into neuronal lineages, produce neurotropic factors, and moderate the immune response makes them valuable tools for considering peripheral neuropathies not just for injuries.

### 4.2. Limitations and Future Directions

The review acknowledges potential limitations, such as variability in study designs, small sample sizes in preclinical studies, and limited clinical trials and long-term clinical outcomes. The success of nerve regeneration is dependent upon a multitude of factors, including the length of the nerve, the location and extent of the injury, the distance from the neuronal cell body to the injury site, the diameter of the nerve and number of axons it contains, whether the axons are myelinated or unmyelinated, the type of nerve fiber (e.g., Aα, Aβ) involved, the direction of transmission (afferent or efferent), and how its electrophysiological propagation is sustained. Additional factors influencing nerve regeneration include the chronicity of the injury, the age of the patient, and the patient’s immune response, which can be affected by comorbidities such as diabetes. Yet, most of these factors were not elaborated. Outcomes should be assessed using standardized methodologies, including electrophysiological nerve conduction studies. These gaps are addressed in the discussion and recommendations for future research. Despite current research, nerve injuries from accidents, infections, diseases, and neuropathy remain a significant problem. Post-COVID, details on an attack on peripheral nerves include Guillain-Barré syndrome, a rare autoimmune disorder triggered by viral infections, causing muscle weakness and loss of reflexes [83]. Stem cells and their exosomes offer potential for systemic and local treatment, including for neuropathies without a known cause.

Refining ADSCs isolation and culture techniques could improve cell potency, scalability, and consistency for clinical use, making them more accessible and effective for nerve regeneration [70]. Next-generation scaffolds that mimic natural nerve architecture, shape memory nanofibers with multichannel conduits, and incorporate bioactive components like ECM proteins and neurotrophic factors could enhance ADSCs-supported nerve repair [84]. Conductive polymers may further support neural signaling in complex injuries. Three-dimensional bioprinting enables the creation of patient-specific customized conduits with embedded ADSCs, tailored to individual nerve injuries [85]. Using genetic engineering to enhance ADSCs’ production of neurotrophic factors (e.g., BDNF, VEGF) could improve survival and accelerate nerve regeneration at injury sites [32]. ADSCs combined with growth factors or anti-inflammatory agents in controlled-release systems could create a more supportive environment for nerve repair [86]. Endoscopic or guided delivery of ADSCs, potentially with injectable hydrogels, could allow for precise, minimally invasive nerve repair. Research on ADSCs’ paracrine and immunomodulatory properties could lead to the development of ADSC-derived secretome therapies, broadening their applications in nerve repair. Long-term clinical trials are essential to establish the safety, optimal dosing, and efficacy of ADSC-based therapies for PNI. Harmonizing regulatory standards and ethical guidelines will facilitate the development and responsible application of ADSC-based therapies. Investigating ADSCs within bioelectronic devices could support neural stimulation, potentially enhancing motor and sensory recovery (Table 4).

Table 4: Future directions for advancing adipose-derived stem cell (ADSC)-based therapies in peripheral nerve repair. Key focus areas include optimizing ADSCs isolation and culture for consistency and scalability, developing advanced scaffolds mimicking extracellular matrix (ECM), and leveraging 3D bioprinting for patient-specific conduits. Genetic modifications aim to enhance ADSCs factor production while combining ADSCs with pharmacological agents can reduce inflammation and support regeneration. Minimally invasive delivery techniques and exploration of paracrine and immunomodulatory effects could broaden applications, including cell-free therapies. Long-term safety trials, regulatory harmonization, and bioelectronic integration represent critical steps toward clinical and commercial translation, particularly for motor and sensory nerve recovery.

## 5. Conclusions

ADSCs are a widely available source of tissue for regenerative purposes, either for autologous or allogeneic purposes. The volume of undifferentiated cells in adipose tissue is comparatively higher than many other sources of tissues in the body. It is easy to harvest and does not need general anesthesia or any sophisticated techniques. Considering their ubiquitous role in regenerative medicine, as it can generate neurotrophic and angiogenic factors, its application in the treatment of PNI is tenable. Though in its early stages, the role of ADSCs in PNI has tremendous potential. Many investigators have published their findings in animal models; however, translation to humans requires more evidence. Clinical trials, standard operating protocols, GMP facilities, appropriate NCs, scaffolds, and universally recognized testing criteria are needed. Despite some shortcomings, ADSCs will be an important source of NTE in the future.

## Figures and Tables

**Figure 1 neurolint-17-00023-f001:**
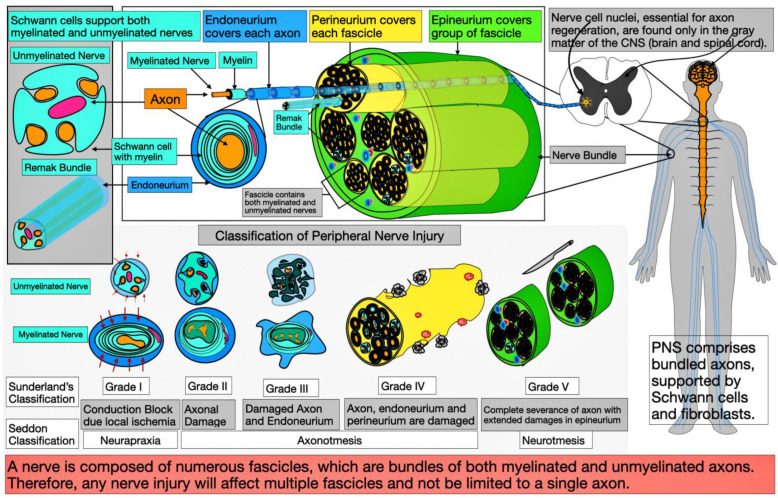
Schematic diagram details the microanatomy and classifications of peripheral nerve injuries (PNI), including both Sunderland and Seddon classifications. The image underscores that the peripheral nervous system (PNS) is not an independent entity and requires the central nervous system (CNS) for repair and regeneration. The intricate network of peripheral nerves, which are structured into fascicles and enclosed within multi-layered shielding sheaths by specialized cells like Schwann cells and fibroblasts, necessitates substantial support for the regenerative process.

**Figure 2 neurolint-17-00023-f002:**
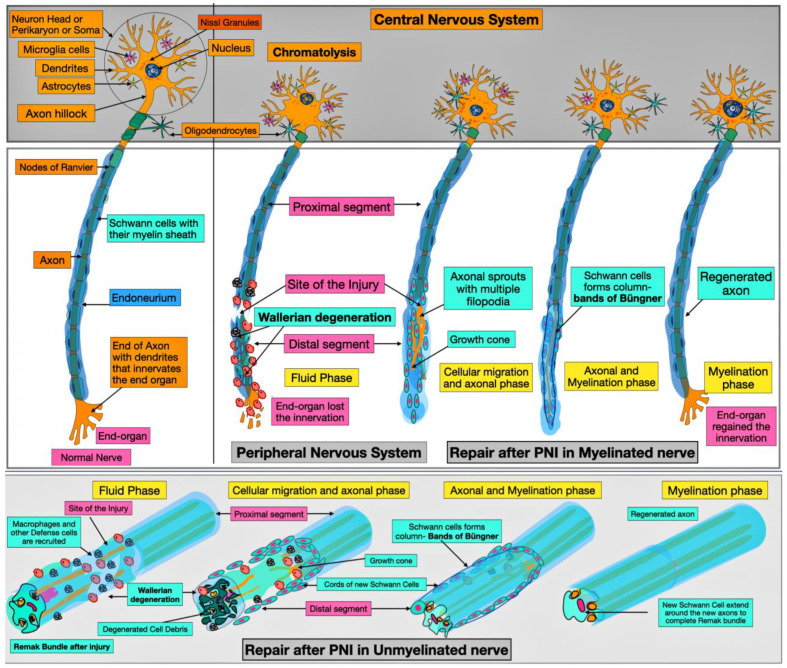
The illustration features Wallerian degeneration at the individual nerve level in both myelinated and unmyelinated nerves. Post-injury, the perikaryon in the CNS will endure chromatolysis: The cell body swells; the nucleus becomes eccentric; and Nissl granules disperse. When needed, microglia cells, astrocytes, and oligodendrocytes are recruited to aid in the recovery of the nerve body. At the PNS level, the injury site divides the axon into a proximal segment, which is continuous with the cell body in the CNS, and a distal segment which is disconnected from the cell body. Wallerian degeneration begins with the fragmented axons initiating modifications in the Schwann cells and recruiting macrophages to remove the damaged myelin and axon debris. These changes are marked by a fluid phase that accumulates defense and scavenger cells around the injured site. Later, during the cellular phase, the Schwann cells guide the sprouting axons by forming bands of Büngner to prune and guide them. The process continues until regeneration is complete.

**Figure 3 neurolint-17-00023-f003:**
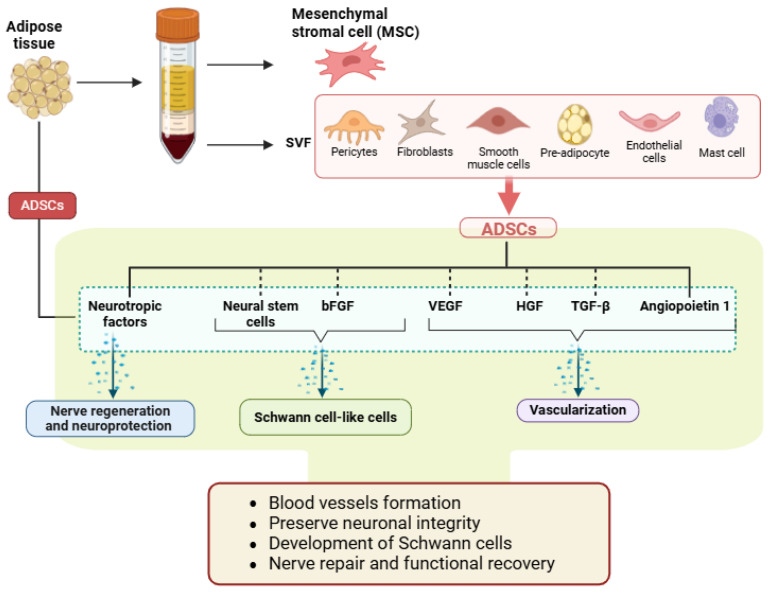
The role of adipose-derived stem cells (ADSCs) in nerve repair and regeneration. ADSCs are derived from adipose tissue and contain mesenchymal stromal cells (MSCs), including fibroblasts, smooth muscle cells, endothelial cells, and mast cells. ADSCs release neurotrophic factors, neural stem cells, and growth factors like bFGF, VEGF, HGF, TGF-β, and Angiopoietin 1, which promote nerve regeneration, neuroprotection, Schwann cell-like differentiation, and vascularization. These mechanisms collectively support blood vessel formation, preservation of neuronal integrity, Schwann cell development, and functional recovery of damaged nerves.

**Figure 4 neurolint-17-00023-f004:**
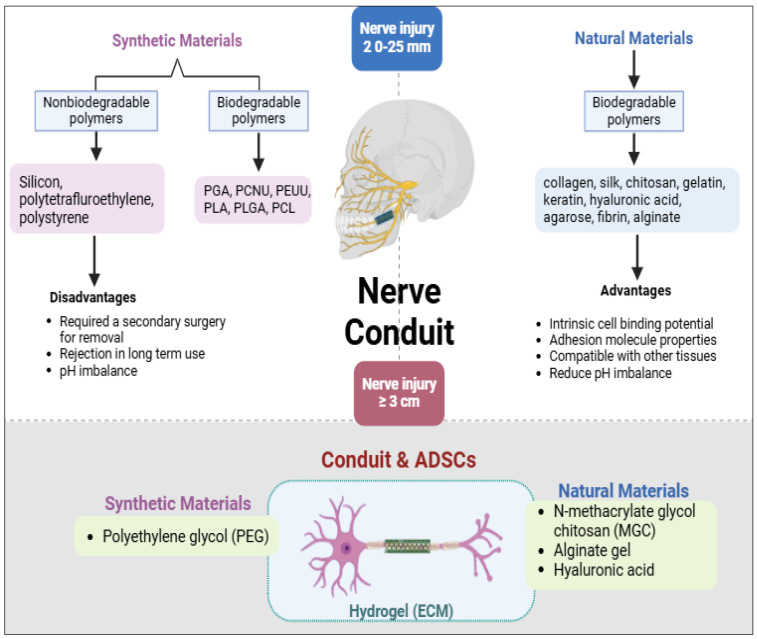
The use of nerve conduits made from synthetic and natural materials for nerve repair based on injury length. Synthetic materials include nonbiodegradable polymers (e.g., silicone, polytetrafluoroethylene) and biodegradable polymers (e.g., PGA, PLA, PCL), offering strength but with disadvantages like the need for secondary surgeries and potential foreign body reactions. Natural materials, such as collagen, gelatin, and alginate, exhibit intrinsic cell adhesion properties and compatibility with tissue repair. For extended injuries, conduits integrated with adipose-derived stem cells (ADSCs) and hydrogels (e.g., polyethylene glycol, chitosan) support regeneration by providing extracellular matrix (ECM) scaffolding, promoting nerve repair through enhanced cell growth and vascularization.

**Table 1 neurolint-17-00023-t001:** Comparison of Treatment Approaches for Nerve Regeneration.

Treatment Approach	Description	Advantages	Limitations	Applications	Reference
**Autologous Nerve Grafting**	Transplanting the patient’s own nerve tissue	Gold standard; integrates well	Donor site morbidity, limited by gap size	Peripheral nerve gaps < 4 cm	[6]
**Stem Cell Therapy (ADSCs)**	Use of ADSCs for nerve repair	High yield, minimally invasive, paracrine effects	Requires further research on long-term safety	Peripheral nerve injuries, small gaps	[7]
**Synthetic Nerve Conduits**	Artificial conduits for guiding nerve regeneration	Available in various materials	May need removal if non-biodegradable; possible rejection	Peripheral nerve gaps, short-to-moderate lengths	[8]
**Natural Nerve Conduits**	Biodegradable conduits, often collagen-based	Biodegradable, cell-binding properties	Limited structural strength, sometimes rapid degradation	Peripheral nerve repair, moderate gaps	[9]

**Table 2 neurolint-17-00023-t002:** Key properties and functions of adipose-derived stem cells (ADSCs).

Property	Description	Impact on Nerve Regeneration	Reference
High Cell Yield	Easily harvested from adipose tissue with minimal invasiveness	Supports scalability for therapeutic use	[36]
Secretion of Growth Factors	Produces growth factors (VEGF, EGF, HGF, IGF1, PGDF, FGF), TGF-β, neurotrophic factors (BDNF, GDNF, NGF)	Promotes angiogenesis and neuroprotection	[7,12,32,43]
Differentiation Potential	Can differentiate into Schwann-like cells and neural cells under appropriate conditions	Supports direct repair and regeneration of nerve tissue	[44]
Paracrine Signaling	Releases bioactive molecules that support nearby cell functions and tissue repair	Enhances local environment for regeneration	[32,45,46,47,48,49]
Immunomodulatory Properties	Modulates immune response to reduce inflammation	Minimizes tissue damage and scarring	[46,49]

**Table 3 neurolint-17-00023-t003:** Scaffold Materials in Nerve Regeneration and Their Interaction with ADSCs.

Material	Type	ADSC Compatibility	Advantages for Nerve Repair	Challenges	Reference
**N-methacrylate glycol chitosan (MGC)**	Hydrogel	High compatibility; injectable	Easy to apply, forms stable cross-linked structures	Needs UV/visible light for cross-linking	[58]
**Alginate Gel**	Natural polysaccharide	Supports ADSC viability, porous	FDA-approved, biodegradable	Forms homogeneous structures	[59]
**Hyaluronic Acid**	Natural polysaccharide	High biocompatibility	Enhances cell migration, suitable for nerve conduits	Rapid degradation in some applications	[60]
**Polyethylene Glycol (PEG)**	Synthetic hydrogel	ECM-similar, supports nutrient transfer	Biocompatible, customizable for nerve tissue	Requires modification for specific uses	[61]

**Table 4 neurolint-17-00023-t004:** Overview of future directions, intended focus, potential impact, and specific applications in the context of ADSCs-based nerve regeneration for PNI.

Future Direction	Focus Area	Potential Impact	Applications	Reference
**Enhanced ADSCs Isolation and Culture**	Optimization of isolation, culture conditions	Increased consistency, potency, and scalability for clinical use	Peripheral nerve repair; large-scale production	[70]
**Advanced Scaffold Development**	Bioengineered, ECM-mimicking scaffolds	Improved support for ADSCs proliferation and nerve integration	Complex nerve injuries; peripheral applications	[84]
**3D Bioprinting for Customized Conduits**	Patient-specific, ADSC-embedded conduits	Precise alignment with individual anatomy, enhanced regeneration	Custom nerve repairs, particularly in orofacial regions	[85]
**Genetic Modification of ADSCs**	Gene editing to enhance factor production	Increased secretion of neurotrophic factors, enhanced survival at injury sites	PNI with high regenerative needs	[32]
**Combination with Pharmacological Agents**	Controlled-release drug delivery within scaffolds	Reduced inflammation, optimized environment for ADSCs	Long-gap or inflamed injury sites in peripheral nerves	[84,86]
**Minimally Invasive Delivery**	Endoscopic or guided ADSCs injection techniques	Reduced trauma and recovery time, precise targeting	Small-to-medium defects, especially in orofacial regions	[33,87]
**Exploring Paracrine and Immunomodulatory Effects**	Study of ADSCs secretome and immune modulation	Development of cell-free therapies, broadening applications	Inflammatory nerve injuries; chronic conditions	[12,32,88]
**Long-term Safety and Efficacy Trials**	Clinical testing for ADSC-based therapies	Evidence-based protocols, ensured safety and efficacy	Peripheral nerve regeneration; regulatory compliance	[89]
**Regulatory and Ethical Considerations**	Harmonization of guidelines for ADSC therapies	Facilitated development, increased public trust	Clinical and commercial applications in nerve repair	[35]
**Bioelectronic Integration**	Combining ADSCs with bioelectronic devices	Enhanced neural stimulation, support for biofeedback systems	Motor and sensory recovery in peripheral nerves	[90]

## Data Availability

No new data were created or analyzed in this study.

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
