# Peer review of "The Potential Role of Adipose-Derived Stem Cells in Regeneration of Peripheral Nerves"

_2035-8377, 2025, doi:10.3390/neurolint17020023_

Round 1

Reviewer 1 Report

Comments and Suggestions for Authors

This manuscript systematically synthesizes existing literature on the role of adipose tissue-derived Stem cells (ADSCs) in peripheral nerve repair and regeneration, employing rigorous methodologies to ensure the inclusion of high-quality, clinically relevant studies. Literature searches were conducted in PubMed, Scopus, and Web of Science using keywords like “adipose tissue-derived stem cells,” “peripheral nerve injury,” and “regenerative medicine,” combined with Boolean operators for refinement. Inclusion criteria focused on peer-reviewed articles, systematic reviews, and clinical trials from 2000 to 2024 while excluding non-peer-reviewed or methodologically weak studies.

Peripheral nerve injuries are frequent complications in surgical and dental practices, often leading to functional impairments and diminished quality of life. Current treatments, such as autografts, face challenges like donor site morbidity and limited efficacy. ADSCs present a promising alternative due to their accessibility, ease of harvesting, and multipotent properties. This manuscript highlights the therapeutic potential of ADSCs, particularly their role in bioengineered nerve conduits and supportive microenvironments. Evidence from case reports, systematic reviews, and clinical trials underscores their ability to promote nerve regeneration through angiogenic and neurotrophic factor secretion, cellular persistence, and scaffold-based tissue repair. ADSCs offer a minimally invasive, autologous option for nerve repair, with future research needed to establish standardized protocols and enhance clinical outcomes, positioning ADSCs as a cornerstone in regenerative medicine.

There are still a few minor issues as follows:

1.    Figure Revision: The first figure seems to overshadow the main content, as the myelinated structure of the peripheral nervous system occupies too much space, making the two types of peripheral nerve injuries less noticeable. It is recommended to adjust this figure for better balance.

2.    Structural Adjustment: The Methodology section (Lines 29–93) could be removed, and the content below can be introduced directly to streamline the flow.

3.    Discussion Section: The current title is not entirely appropriate since this is not an experimental study. As a review article, it would be better to build upon the literature and present original perspectives.

4.    Literature Focus: Greater emphasis should be placed on the potential role of adipose-derived stem cells, ensuring that this aspect is highlighted more prominently in the references.

Author Response

We sincerely appreciate your constructive feedback. Your comments have been invaluable in refining the manuscript, improving its flow, and reducing repetition. These changes will undoubtedly enhance reader engagement and comprehension.

Comment 1: Figure Revision: The first figure seems to overshadow the main content, as the myelinated structure of the peripheral nervous system occupies too much space, making the two types of peripheral nerve injuries less noticeable. It is recommended to adjust this figure for better balance.

Response 1: Thank you for bringing this important detail to our attention. The necessary changes have been implemented in Figure 1. It was challenging to find details regarding the repair of unmyelinated nerves after injury, and the necessary information has been added. Figure 2 was also adjusted to enhance the presentation of both myelinated and unmyelinated nerves.

Comment 2: Structural Adjustment: The Methodology section (Lines 29–93) could be removed, and the content below can be introduced directly to streamline the flow.

Response 2: Thank you for your feedback on improving the flow of the manuscript. The suggested lines have been restructured.

Comment 3: Discussion Section: The current title is not entirely appropriate since this is not an experimental study. As a review article, it would be better to build upon the literature and present original perspectives.

Response 3: Thank you for your feedback. The discussion section of the article has been improved by including literature details on adipose-derived stem cells.

Comment 4: Literature Focus: Greater emphasis should be placed on the potential role of adipose-derived stem cells, ensuring that this aspect is highlighted more prominently in the references.

Response 4: Thank you for your constructive feedback. We have incorporated additional details regarding adipose-derived stem cells to enhance the clarity of the manuscript.

Reviewer 2 Report

Comments and Suggestions for Authors

Perepheral nerve injuries (PNI) are common in clinic and adipose derived stem cells (ADSCs) have been shown with important application in the therapy of PNI. The manuscript systematically reviewed the progress and problems in this field. Overall it's well written review and will provide essential reference for the related scientists. A few minor issues need to be improved are listed here.

1. In the top left corner of Fig 1, "Schwann cells support both myelinated and unmyelinated nerves" should be corrected. Because Schwann cells only support myelination in peripheral nervous system (PNS). In central nervous system (CNS), however, it is oligodendrocytes to form the myelin sheath. Please also note the term "peripheral nervous system/peripheral nerves system", it's should be correct and consistent from the title to the end of the manuscript.

2. In top left corner of Fig 2, the black arrow for "Neuron Head or Perikaryon or Soma" is not at the right place, please correct.

3. For the definition and application  of abbreviations, please define them when they first appear and then use the abbreviations thereafter. The whole manuscript will be concise in this way. Check all the abbreviations from the begaining to the end, including the figure legends, table titles and descriptions. 

Author Response

We sincerely appreciate your constructive feedback. Your comments have been invaluable in refining the manuscript, improving its flow, and reducing repetition. These changes will undoubtedly enhance reader engagement and comprehension.

Commnet1: In the top left corner of Fig 1, "Schwann cells support both myelinated and unmyelinated nerves" should be corrected. Because Schwann cells only support myelination in peripheral nervous system (PNS). In central nervous system (CNS), however, it is oligodendrocytes to form the myelin sheath. Please also note the term "peripheral nervous system/peripheral nerves system", it's should be correct and consistent from the title to the end of the manuscript.

Response 1: Thank you for your thorough review and for indicating the necessary changes to strengthen the manuscript. As per your suggestion, Figure 1 and 2 has been modified to include oligodendrocytes for the CNS, in addition to the Schwann cells for the PNS. Furthermore, the term "peripheral nervous system" has been used consistently throughout the manuscript.

Comment 2: In top left corner of Fig 2, the black arrow for "Neuron Head or Perikaryon or Soma" is not at the right place, please correct.

Response 2: Thank you for bringing this minor detail to our attention. The necessary changes have been implemented in Figure 2.

Comment 3: For the definition and application  of abbreviations, please define them when they first appear and then use the abbreviations thereafter. The whole manuscript will be concise in this way. Check all the abbreviations from the begaining to the end, including the figure legends, table titles and descriptions. 

Response 3: Your attention to detail in maintaining consistency throughout the manuscript is highly appreciated. The abbreviations, figure legends, table titles, and descriptions have been adjusted to ensure a professional presentation. This will enhance the overall quality of our manuscript.

Reviewer 3 Report

Comments and Suggestions for Authors

The manuscript explores the therapeutic potential of adipose-derived stem cells (ADSCs) in regenerating peripheral nerves. However, before the manuscript can be accepted, the following recommendations should be addressed: 

1.  The headline "2.1. The Potential Role of Adipose Derived Stem Cells in Regeneration of Peripheral Nerves" is not appropriate directly under the main headline "2. Methodology" because it does not align with the section's focus.

2. Consolidate repetitive discussions, such as the secretion of neurotrophic factors by ADSCs, into a single, detailed section. Avoid reintroducing the same concepts across multiple parts of the manuscript.

3. Please address challenges in ADSC therapies, such as scalability, ethical concerns, and variability in clinical outcomes, with more depth and supporting references.

4. Please revise the manuscript for grammatical accuracy, consistent tense usage, and typographical errors to ensure professionalism.

Author Response

We sincerely appreciate your constructive feedback. Your comments have been invaluable in refining the manuscript, improving its flow, and reducing repetition. These changes will undoubtedly enhance reader engagement and comprehension.

Comment 1:  The headline "2.1. The Potential Role of Adipose Derived Stem Cells in Regeneration of Peripheral Nerves" is not appropriate directly under the main headline "2. Methodology" because it does not align with the section's focus.

Response 1: Thank you for highlighting the disrupted flow within the manuscript. Your comment is greatly appreciated, as it has significantly improved the overall structure and readability. The section has been restructured under a separate heading and is now presented separately from the methodology section as follows: 

  1. The Pathophysiological Cascade Following Peripheral Nerve Injury

Comment 2: Consolidate repetitive discussions, such as the secretion of neurotrophic factors by ADSCs, into a single, detailed section. Avoid reintroducing the same concepts across multiple parts of the manuscript. 

Response 2: Thank you for bringing the repetitive presentation to our attention. We have carefully reviewed the manuscript and consolidated relevant information into a single section, while simplifying other areas to maintain a clear and concise flow.

Comment 3: Please address challenges in ADSC therapies, such as scalability, ethical concerns, and variability in clinical outcomes, with more depth and supporting references.

Response 3: As per your guidance, we have incorporated references and points related to scalability, ethical concerns, and variability in clinical outcomes. These additions are now displayed in the manuscript.

Comment 4: Please revise the manuscript for grammatical accuracy, consistent tense usage, and typographical errors to ensure professionalism.

Response 4: The manuscript has been thoroughly revised for grammatical accuracy. Thank you for identifying the errors, which has enhanced the overall professionalism of the presentation. The manuscript was comprehensively restructured to enhance grammatical accuracy and clarity.